# Immunohistochemical Features of Indoleamine 2,3-Dioxygenase (IDO) in Various Types of Lymphoma: A Single Center Experience

**DOI:** 10.3390/diagnostics10050275

**Published:** 2020-05-02

**Authors:** Mee-seon Kim, Tae In Park, Shin-Ah Son, Hyoun Wook Lee

**Affiliations:** 1Department of Pathology, Kyungpook National University Hospital, 130 Dongdeok-ro, Jung-gu, Daegu 41944, Korea; kimm2342@gmail.com; 2Department of Pathology, Kyungpook National University School of Medicine, 680 Gukchaebosang-ro, Jung-gu, Daegu 41944, Korea; tipark@knu.ac.kr; 3Department of Thoracic and Cardiovascular Surgery, Kyungpook National University Hospital, 130 Dongdeok-ro, Jung-gu, Daegu 41944, Korea; sina2-2@daum.net; 4Department of Pathology, Samsung Changwon Hospital, Sungkyunkwan University School of Medicine, Changwon 51353, Korea

**Keywords:** Indolamine-2,3-dioxygenase (IDO), lymphoma, immunohistochemistry (IHC)

## Abstract

Indolamine-2,3-dioxygenase (IDO) is an intracellular enzyme that catalyzes amino acid tryptophan to L-kynurenine. IDO is overexpressed in various cancers and several IDO inhibitors have been assessed in multiple clinical trials. If an IDO inhibitor is to be commercialized, IDO immunohistochemistry will be an important method. In this study, 80% (28/35) of mature T- and natural killer (NK)-cell neoplasms showed positivity for IDO protein (score 1: five, score 2: one, score 3: seven, score 4: fifteen). In addition, 29.9% (23/77) of mature B-cell lymphomas showed positivity for IDO protein (score 1: three, score 2: tewelve, score 3: four, score 4: four). In mature B-cell lymphomas, 95.7% (22/23) of IDO positive cases were diffuse B-cell lymphomas. Our study includes various types of lymphoma that were previously unreported and shows various patterns of IDO stain according to the type. When the results are accumulated, IDO immunohistochemistry will be a useful tool to diagnose lymphomas and to predict their prognosis.

## 1. Introduction

Tumors express the antigens that induce the host immune response. Progression of tumors requires avoidance of host immune surveillance [1,2]. Recent studies have shown that tryptophan catabolism is one means of avoiding immune surveillance [3,4]. Indolamine-2,3-dioxygenase (IDO) is a cytosolic enzyme that catalyzes tryptophan. IDO converts the amino acid tryptophan to L-kynurenine [5]. The depletion of tryptophan and the production of L-kynurenine induces the apoptosis of T-cells and natural killer (NK)-cells [6,7,8]. In addition, the IDO-expressing macrophages, dendritic cells, and tumor cells suppress T-cell proliferation [7,8,9,10]. In previous reports, IDO expression and the serum concentration of L-kynurenine were negative prognostic factors in diffuse large B-cell lymphomas and adult T-cell leukemia/lymphomas [11,12,13]. In a previous immunohistochemical analysis for IDO expression in diffuse large B-cell lymphomas treated with R-CHOP chemotherapy, the IDO-positive group showed resistance to the treatment and a poorer prognosis than the IDO-negative group [14]. Immunohistochemistry is a relatively fast and inexpensive utility in diagnostic surgical pathology. Immunohistochemistry is widely used for subtyping of lymphomas and plays an important role in hematopathology. There are very few recent immunohistochemical assays of IDO in lymphomas [14,15,16]. To address different immunohistochemical features in various lymphomas, we performed immunohistochemistry of IDO in a Korean lymphoma cohort of a single center.

## 2. Materials and Methods

### 2.1. Study Population

This study was approved by the Institutional Review Board (IRB) of Samsung Changwon Hospital, Changwon, Korea (IRB FILE No. 2020-01-003, 23 January 2020). The study was retrospective, therefore the IRB waived the need for written informed consent. The medical records of Samsung Changwon Hospital between January 2014 and December 2019 were gathered. All slides of diagnosed lymphomas during the period were independently reviewed by two authors (H.Y.L and T.I.P) according to the World Health Organization (WHO) classification of tumors of hematopoietic and lymphoid tissues, 4th Edition. Of a total of 171 cases obtained by biopsy or excision, those with an insufficient amount of specimen (cut off: 0.25 cm²) and cases of controversial diagnosis were excluded from the study. The remaining 120 cases were enrolled in this study (Male:Female = 5:3; aged 10–86, mean = 59.4 years, median = 62 years). Of the 120 cases of lymphoma, 103 cases were Ann Arbor stage I, 12 cases were stage II, and five cases were stage III. In situ hybridization (ISH) with the Epstein-Barr virus (EBV)-encoded small RNA (EBER) were performed in 91 cases of lymphoma. A total of 26.4% (24/91) of cases showed positivity for ISH with EBER (Hodgkin Lymphoma: five, EBV-Positive diffuse large B-cell lymphoma (DLBCL), not otherwise specified (NOS): two, Extranodal NK-/T-cell Lymphoma: twelve, Peripheral T-cell Lymphoma, NOS: three, Angioimmunoblastic T-cell Lymphoma: one, Enteropathy-associated T-cell Lymphoma, Type II: one). All cases were negative for HIV infection. All specimens were obtained at the time of pathologic diagnosis before initiation of treatment. A total of seven cases of Hodgkin lymphoma, 77 cases of mature B-cell lymphoma, one B-Lymphoblastic lymphoma, and 35 cases of mature T- and NK-cell neoplasm were enrolled the study.

### 2.2. Immunohistochemistry for Indoleamine 2, 3-Dioxygenase

We reviewed all slides of the cases and selected one representative formalin-fixed, paraffin-embedded (FFPE) block from each case for immunohistochemistry. The representative blocks were cut on 4 μm thick sections and immunohistochemical staining was performed for Indoleamine 2, 3-dioxygenase (rabbit recombinant monoclonal, clone EPR20374, Abcam, 1:2000 dilution). All immunohistochemical staining was performed using the BenchMark XT autostainer (Ventana Medical Systems, Tucson, Arizona), according to the manufacturer’s protocol. The results were evaluated by two authors (H.Y.L and M.S.K) independently, and any discrepancies were reviewed to achieve a consensus. In previous studies, the reactive immune cells around the tumors were stained by IDO protein [17,18]. It was difficult or impossible to distinguish reactive immune cells from tumor cells. Considering the possibility of containing reactive cells in the tumors, percentage of stained cells equal to or higher than 5% were regards as positive. All lymphomas were scored based on percentage of tumor cell staining: 0 = <5%, 1 = 5–25%, 2 = 26–50%, 3 = 51–75%, and 4 = >75%. The moderate-to-strong cytoplasmic staining was regards as positive. In Hodgkin lymphomas, due to the scant cellularity of tumor cells, the expressions were evaluated to be positive or negative. The positivity for IDO in the vessels was judged in comparison with hematoxylin and eosin (H&E) stain. The vessels were lined by endothelium and covered by pericytes in H&E stain.

## 3. Results

### 3.1. Hodgkin Lymphoma

The clinicopathologic characteristics and the immunohistochemical results of IDO are summarized in Table 1. The male:female ratio was 1.3:1, and the age distribution was 35–79 years (mean = 61 years, median = 70 years). Four cases of classic Hodgkin lymphoma (three mixed cellularity and one nodular sclerosis), and three cases of nodular lymphocyte predominant Hodgkin lymphoma were included in the study. There was no difference of IDO staining pattern between classic type and nodular lymphocyte predominant type. In the positive cases, the tumor cells and vessels showed positivity for IDO protein (Figure 1). In the negative cases, only dendritic cells and macrophages showed positivity for IDO (Figure 2). Three cases were positive, and four cases were negative. In past studies [15,16], IDO had been expressed in the dendritic cells and the macrophages only. However, our results showed tumor cell positivity for IDO in several cases.

### 3.2. Diffuse Large B-Cell Lymphoma (DLBCL)

#### 3.2.1. Diffuse Large B-Cell Lymphoma (DLBCL), NOS

A total of 26 cases of diffuse large B-cell lymphoma, NOS were enrolled the study. The clinicopathologic characteristics and the immunohistochemical results of IDO are summarized in Table 2. The male:female ratio was 2.7:1, and the age distribution was 10–84 years (mean = 61 years, median = 65.5 years). Six cases were germinal center type and 20 cases were activated B-cell type. There was no difference in the IDO staining pattern between germinal center type and activated B-cell type. The lymphomas were scored based on percentage of tumor cell staining: 0 = <5%, 1 = 5–25%, 2 = 26–50%, 3 = 51–75%, and 4 = >75% (Figure 3A: score 0; 3B: 1; 3C: 2; 3D: 3; 3E: 4). The staining distribution was scored from 0 to 4. Eight cases were scored 0, three cases were 1, 10 cases were 2, three cases were 3, and two cases were 4.

#### 3.2.2. Diffuse Large B-Cell Lymphoma (DLBCL), Subtypes

The clinicopathologic characteristics and the immunohistochemical results of IDO are summarized in Table 2. A total of 30.8% (8/26) of DLBCL, NOS were scored 0 for IDO immunohistochemistry, but four cases of primary DLBCL of the central nervous system were scored 0 (Figure 4). A T-cell/histiocyte–rich large B-cell lymphoma was scored 2 (Figure 3F), a primary cutaneous DLBCL, leg type was scored 4 (Figure 3G), and one case of EBV-positive DLBCL, NOS was scored 2 and another was scored 4 (Figure 3H,I).

### 3.3. B-Cell Lymphomas with Low IDO Expression

A total of 15 cases of follicular lymphoma (grade 1–2: 11, 3A: three, 3B: one) and 16 cases of marginal zone B-cell lymphoma of the mucosa-associated lymphoid tissue (MALT lymphoma) were enrolled in the study. The clinicopathologic characteristics and the immunohistochemical results of IDO are summarized in Table 3. The male:female ratio was 1.14:1, and the age distribution was 36–79 years (mean = 52.3 years, median = 50 years) in follicular lymphomas. The male:female ratio was 1:1, and the age distribution was 42–84 years (mean = 64.1 years, median = 64 years) in MALT lymphomas. A total of 93.3% (14/15) of follicular lymphomas were scored 0 (Figure 5A). One 3B case showed positivity for IDO in the centroblasts of follicles and the diffuse patterned areas (Figure 5B, submandibular lymph node and, 16 months later, the lymphoma transformed to DLBCL of the brain (IDO score 4). One pediatric follicular lymphoma was scored 0 for IDO expression. A total of 16 cases of MALT lymphoma (Figure 5C) and four cases of nodal marginal zone lymphoma were scored 0 for IDO. Three mantle lymphomas (Figure 5D), two chronic lymphocytic leukemia/small lymphocytic lymphomas (Figure 5E), and one B-lymphoblastic lymphoma/leukemia, NOS (Figure 5F) were scored 0 for IDO expression. Two Burkitt lymphomas were scored 0 for IDO (Figure 6).

### 3.4. Mature T-and NK-Cell Neoplasms

Twelve cases of extranodal NK-/T-cell lymphoma (aged 19–83 (mean = 54.2, median = 57), Male:Female = 1.4:1), eight cases of primary cutaneous CD30 positive T-cell proliferative disorder (six lymphomatoid papulosis, two primary cutaneous anaplastic large cell lymphoma), six cases of peripheral T-cell lymphoma, NOS, five cases of anaplastic large cell lymphoma (three ALK-positive and two ALK-negative), a primary cutaneous CD8-positive aggressive epidermotrophic cytotoxic T-cell lymphoma, a enteropathy-associated T-cell lymphoma, a angioimmunoblastic T-cell lymphoma, and a subcutaneous panniculitis-like T-cell lymphoma were enrolled in the study. The clinicopathologic characteristics and the immunohistochemical results of IDO are summarized in Table 4. Of the 12 cases of extranodal NK-/T-cell lymphoma, 10 cases showed diffuse positivity for IDO protein (Figure 7A) and only two cases were scored 0 (Figure 7B). The IDO negative cases were younger (19 and 26 years) than IDO positive cases (aged 43–83). The staining distribution of IDO in lymphomatoid papulosis was scored 0–4 (scored 0: one, scored 1: three, scored 4: two; Figure 8A–D). One case of primary cutaneous anaplastic large cell lymphoma was scored 3 (Figure 8E,F) and another was scored 0. The staining distribution of IDO in peripheral T-cell lymphoma, NOS was scored 1–3 (score 1: one, Figure 9A,B; score 3: 5, Figure 9C,D). The staining distribution of IDO in anaplastic large cell lymphoma, ALK(anaplastic lymphoma kinase)-positive was scored 2–4 (score 2: one, score 3: one, score 4: one, Figure 10A,B). All cases of anaplastic large cell lymphoma, ALK-negative were scored 0 (Figure 10C,D). Primary cutaneous CD8-positive aggressive epidermotrophic cytotoxic T-cell lymphoma was scored 4 (Figure 11A,B). Enteropathy-associated T-cell lymphoma, type II was scored 0 (Figure 11C and D). Angioimmunoblastic T-cell lymphoma was scored 1 (Figure 11E), and subcutaneous panniculitis-like T-cell lymphoma was scored 4 (Figure 11F).

## 4. Discussion

Indolamine-2,3-dioxygenase (IDO) is an intracellular enzyme that catalyzes amino acid tryptophan to L-kynurenine [19]. IDO is reportedly expressed by dendritic cells in tumor-draining lymph nodes [10]. In our study, dendritic cells and macrophages of lymph nodes or tonsils showed positivity for IDO protein. All lymphomas were scored based on percentage of tumor cell staining: score 0 = <5%, 1 = 5%–25%, 2 = 26%–50%, 3 = 51%–75%, and 4 = >75%. It was difficult or impossible to distinguish reactive immune cells from tumor cells. To clarify the distinction between the tumor cells and reactive cells, multiple immunofluorescence stainings against IDO and other antibodies can be helpful as has been done in previous research [17]. The staining distribution of IDO was scored 0–4. A total of 80% (28/35) of mature T- and NK-cell neoplasms showed positivity for IDO protein (score 1: five, score 2: one, score 3: seven, score 4: 15). There was no different staining pattern or positive rate between the histopathological types of mature T- and NK-cell neoplasms. A total of 78.6% (22/28) of positive cases were scored 3 or 4; 83.3% (10/12) of extranodal NK-/T-cell lymphomas showed diffuse positivity for IDO protein. The remaining two negative cases were of a relatively young age (19 and 26 years) than the positive extranodal NK-/T-cell lymphomas (aged 43–83). In anaplastic large cell lymphomas, all three ALK-positive cases were positive (scoring 2, 3, or 4) for IDO and both ALK-negative cases were negative for IDO. A total of six peripheral T-cell lymphomas, NOS showed positivity for IDO (five scoring 3 and one scoring 1). In contrast to mature T- and NK-cell neoplasms, 29.9% (23/77) of mature B-cell lymphomas showed positivity for IDO protein (score 1: three, score 2: 12, score 3: four, score 4: four). In mature B-cell lymphomas, 95.7% (22/23) of IDO positive cases were DLBCL, NOS, or DLBCL subtypes. There was no different staining pattern or positive rate between DLBCL, NOS, and DLBCL subtypes. All small B-cell neoplasms except one were negative for IDO protein. The remaining positive case was follicular lymphoma, grade 3B. The positive follicular lymphoma was scored 3 at centroblasts in follicle and diffuse areas. In contrast to other subtypes of DLBCL, all four primary DLBCLs of the CNS were negative for IDO protein. IDO proteins have been overexpressed in various cancers [20,21,22,23] and several IDO inhibitors have been assessed in multiple clinical trials. If an IDO inhibitor is to be commercialized, IDO immunohistochemistry will be important method. However, there are insufficient studies of immunohistochemistry for IDO in lymphomas. Previous studies have demonstrated only Hodgkin lymphoma and diffuse large B-cell lymphoma [14,15,16]. There was no immunohistochemical study of IDO protein for mature T- and NK-cell neoplasms. However, our study included various types of mature T- and NK-cell lymphoma (that have previously been unreported) and showed various patterns of IDO staining according to the type. Our study used scant sample size to determine the IDO pattern of various lymphomas and the relationship between IDO expression and prognosis. When the results are accumulated, IDO immunohistochemistry will be a useful tool to diagnose lymphomas and predict their prognosis.

## Figures and Tables

**Figure 1 diagnostics-10-00275-f001:**
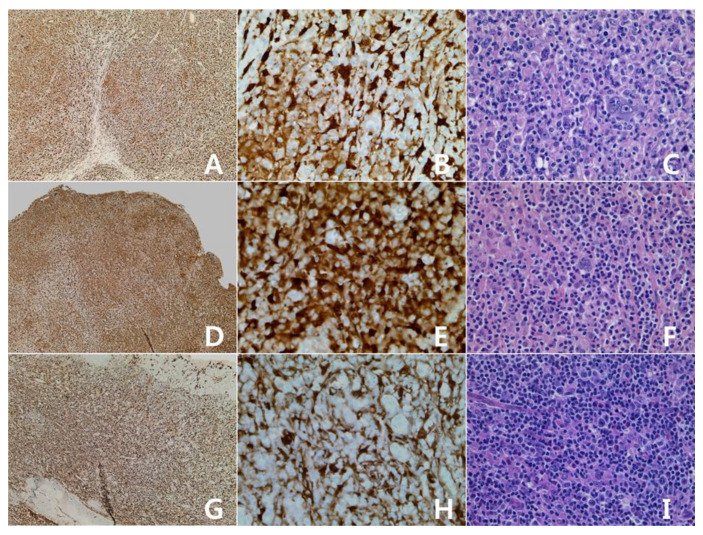
Hodgkin lymphomas with expression of Indolamine-2,3-dioxygenase (IDO): classic Hodgkin lymphoma, mixed cellularity. (**A**–**C**): cervical lymph node; (**G**–**I**): palatine tonsil. Some tumor cells and vascular structures show reactivity for IDO protein (**A**,**G**): IDO stain, ×40; (**B**,**H**): IDO stain, ×400; (**C**,**I**): H&E stain, ×400. Nodular lymphocyte predominant Hodgkin lymphoma (**D**–**F**): cervical lymph node. Some tumor cells and vascular structures show reactivity for IDO protein (**D**): IDO stain, ×40; E: IDO stain, ×400; F: H&E stain, ×400).

**Figure 2 diagnostics-10-00275-f002:**
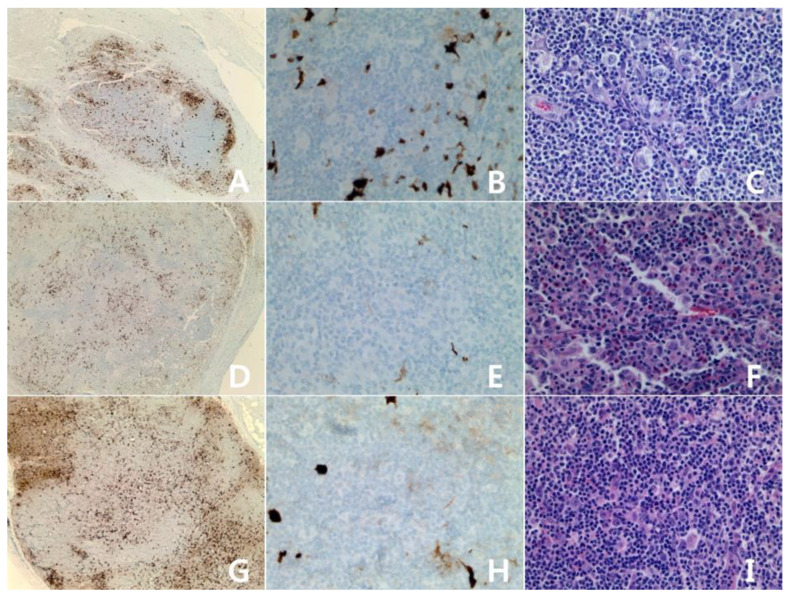
Hodgkin lymphomas, negative for IDO: Classic Hodgkin lymphoma, nodular sclerosis type (**A**–**C**): axillary lymph node. Classic Hodgkin lymphoma, mixed cellularity (**D**–**F**): cervical lymph node. Nodular lymphocyte predominant Hodgkin lymphoma (**G**–**I**): cervical lymph node. Some dendritic cells and macrophages show reactivity for IDO protein (**A**,**D**,**G**): IDO stain, ×40; (**B**,**E**,**H**): IDO stain, ×400: (**C**,**F**,**I**): H&E stain, ×400.

**Figure 3 diagnostics-10-00275-f003:**
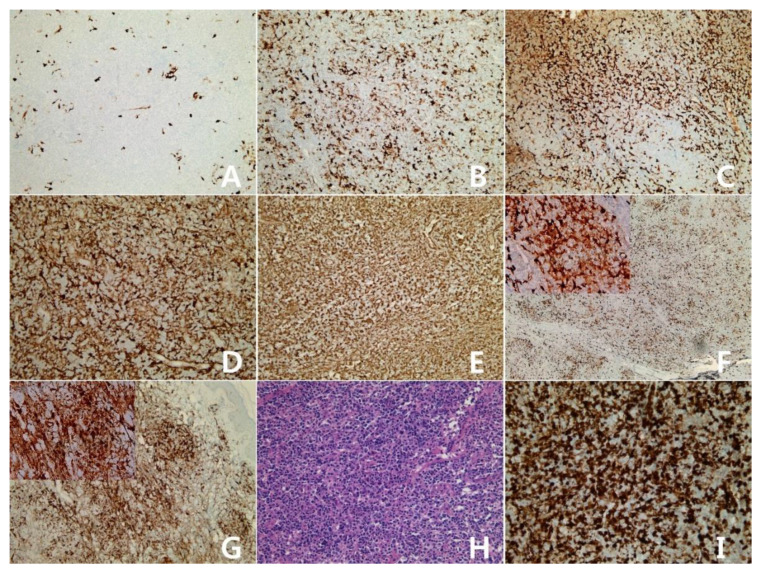
**Diffuse large B-cell lymphoma** with expression of IDO: Diffuse large B-cell lymphoma, not otherwise specified(**A**–**E**), ×100; (**A**): score 0 (<5%), (**B**): score 1 (5-25%), (**C**): score 2 (26–50%), (**D**): score 3 (51-75%), (**E**): score 4 (>75%)). T-cell/histiocyte–rich large B-cell lymphoma (**F**): score 2, ×40; Insert: ×400. Primary cutaneous DLBCL, leg type (**G**): score 4, ×100; Insert: ×400. Epstein-Barr virus (EBV) -positive DLBCL, NOS (**H**): H&E stain, (**I**): IDO stain, score 4, (**H**,**I**): ×200.

**Figure 4 diagnostics-10-00275-f004:**
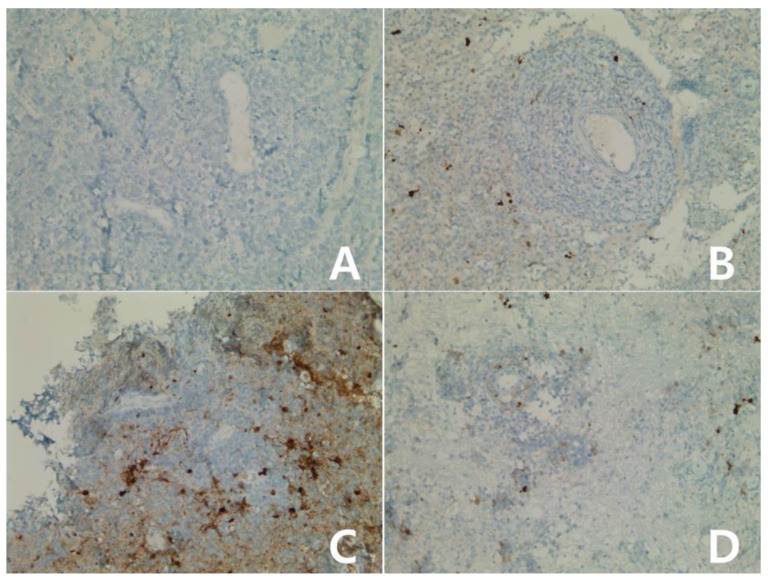
Immunohistochemical stain for IDO in primary DLBCL of the CNS (central nervous system): Atypical lymphocytic infiltration mainly in the perivascular area without IDO expression (**A**–**D**): score 0, ×400).

**Figure 5 diagnostics-10-00275-f005:**
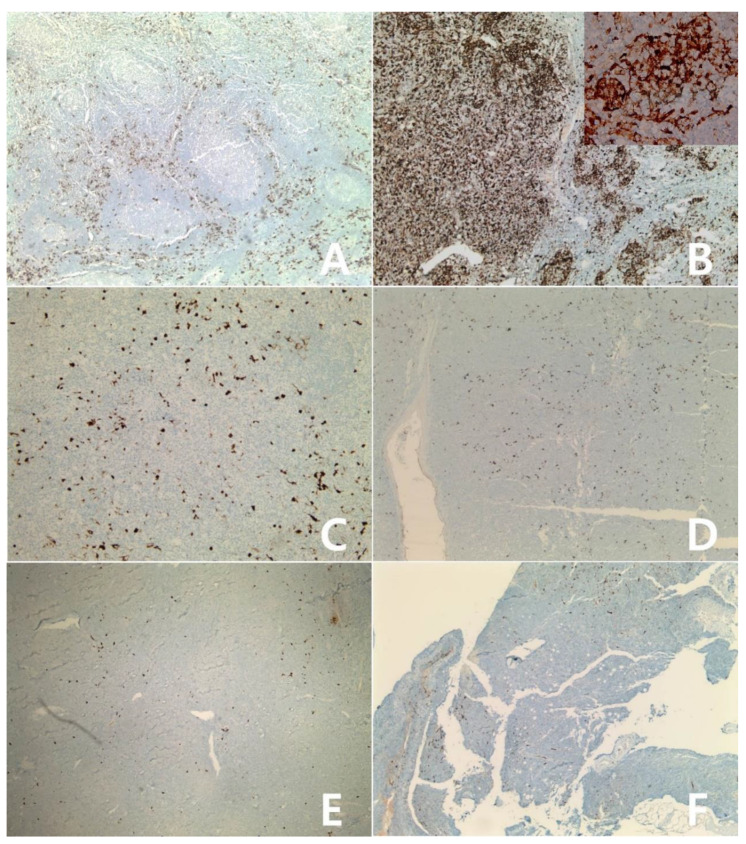
B-cell lymphomas with low IDO expression: Follicular lymphoma, grade 1–2 (**A**), ×40. Follicular lymphoma, grade 3B showed positivity in the centroblasts of follicles and the diffuse patterned areas (**B**), ×40; Insert: ×400. MALT lymphoma (**C**), ×100. Mantle cell lymphoma (**D**), ×40. Chronic lymphocytic leukemia/small lymphocytic lymphoma (**E**), ×40. B-lymphoblastic lymphoma/leukemia (**F**), ×40.

**Figure 6 diagnostics-10-00275-f006:**
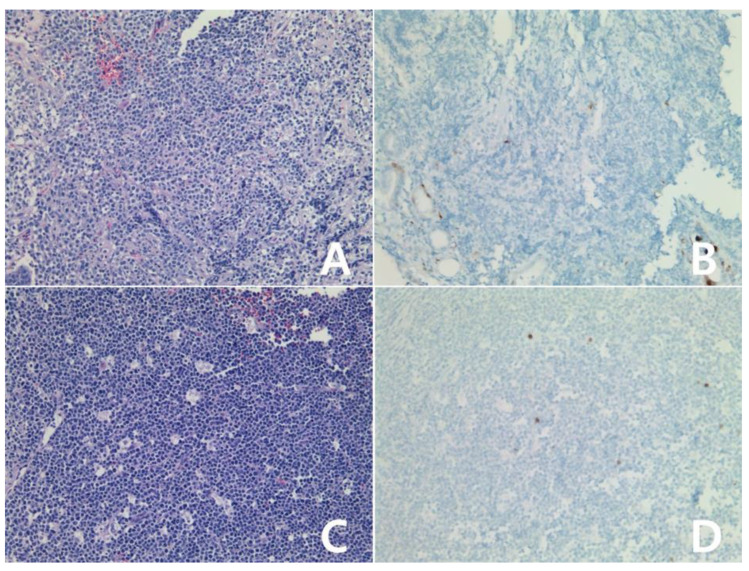
Immunohistochemical stain for IDO in Burkitt lymphoma: Atypical lymphocytes showed negativity for IDO protein (**A**,**B**): case 1, (**C**,**D**): case 2, (**B**,**D**): IDO stain, (**A**,**C**): H&E, all ×200.

**Figure 7 diagnostics-10-00275-f007:**
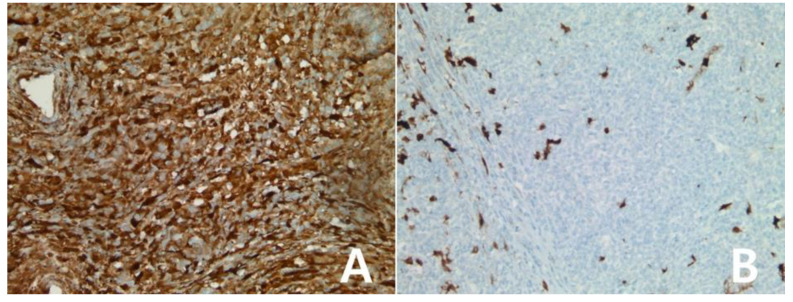
Immunohistochemical stain for IDO in extranodal NK-/T-cell lymphoma: The tumor cells showed diffuse positivity for IDO protein (**A**), ×200. The tumor was scored 0 for IDO protein (**B**), ×200.

**Figure 8 diagnostics-10-00275-f008:**
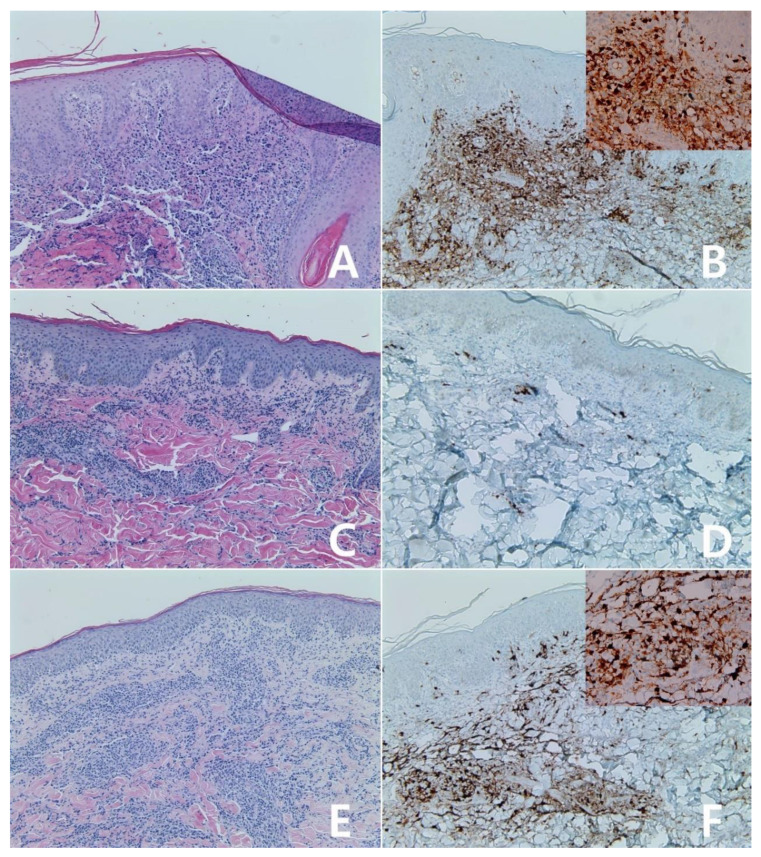
Immunohistochemistry for IDO in primary cutaneous CD30 positive T-cell proliferative disorder: Lymphomatoid papulosis (**A**,**B**): case 1, (**C**,**D**): case 2, (**A**–**D**) ×100, (**A**,**C**): H&E stain. Atypical lymphocytes showed diffuse positivity for IDO protein (**B**), score 4; Insert: ×400, negative for IDO (**D**), score 0. One primary cutaneous anaplastic large cell lymphoma was scored 3 (**E**,**F**), ×100; Insert: ×400, (**E**): H&E stain.

**Figure 9 diagnostics-10-00275-f009:**
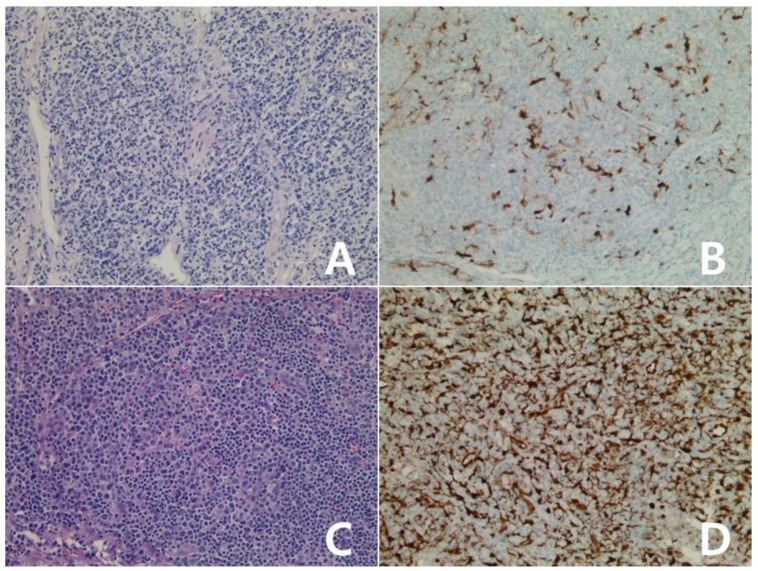
Immunohistochemistry for IDO in peripheral T-cell lymphoma, NOS: (**A**,**B**): scored 1, (**C**,**D**): scored 3, (**A**–**D**) ×200, (**A**,**C**): H&E stain. The tumor was scored 1 for IDO (**B**). The tumor cells showed positivity for IDO, score 3 (**D**).

**Figure 10 diagnostics-10-00275-f010:**
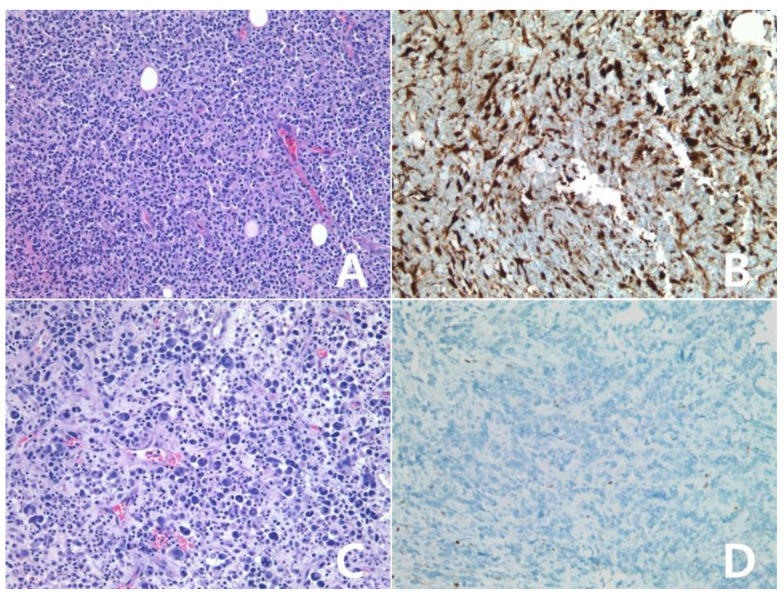
Immunohistochemistry for IDO in anaplastic large cell lymphoma: Anaplastic large cell lymphoma, ALK-positive (**A**,**B**), ×200, A: H&E stain. The tumor showed score 2 for IDO protein (**B**). Anaplastic large cell lymphoma, ALK-negative (**C**,**D**), ×200, C: H&E stain. The tumor showed no reactivity for IDO protein, score 0 (**D**).

**Figure 11 diagnostics-10-00275-f011:**
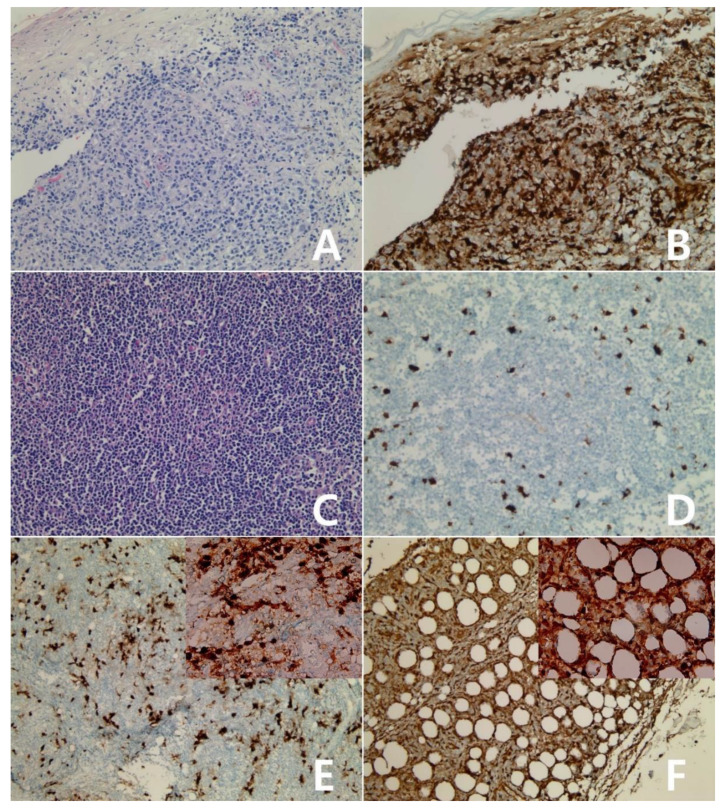
Immunohistochemistry for IDO in mature T- and NK-cell neoplasms: Primary cutaneous CD8-positive aggressive epidermotrophic cytotoxic T-cell lymphoma (**A**,**B**), ×200, A: H&E stain. The tumor showed diffuse reactivity for IDO protein, score 4 (**B**). Enteropathy-associated T-cell lymphoma, type II (**C**,**D**), ×200, C: H&E stain. The tumor showed weak reactivity for IDO protein, score 0 (**D**). Angioimmunoblastic T-cell lymphoma(E), ×100; Insert: ×400. The tumor was scored 1 for IDO protein (**E**). Subcutaneous panniculitis-like T-cell lymphoma (**F**), ×100; Insert: ×400. The tumor showed diffuse reactivity for IDO protein, score 4 (**F**).

**Table 1 diagnostics-10-00275-t001:** The clinicopathologic characteristics and the Indolamine-2,3-dioxygenase (IDO) expressions of Hodgkin lymphomas.

Histologic Type	Age	Sex	Site of Involvement	IDO	
Classic Hodgkin Lymphoma				Tumor Cell	Vessel
**Mixed Cellularity**	70	M	Cervical lymph node	Positive	Positive
	35	F	Cervical lymph node	Negative	Negative
	79	F	Palatine tonsil	Positive	Positive
**Nodular Sclerosis**	72	M	Axillary lymph node	Negative	Negative
**Nodular Lymphocyte Predominant**	33	M	Cervical lymph node	Negative	Negative
	77	M	Cervical lymph node	Positive	Positive
	61	F	Cervical lymph node	Negative	Negative

**Table 2 diagnostics-10-00275-t002:** The clinicopathologic characteristics and the IDO expressions in diffuse large B-cell lymphomas.

Histologic Type	Age	Sex	Site of Involvement	IDO
Diffuse Large B-cell Lymphoma, NOS				
**Germinal Center B-cell**	42	M	Nasal cavity	score 3
	65	M	Nasal cavity	score 1
	10	M	Uvula, palatine tonsil	score 2
	81	F	Femur, brain	score 0
	59	F	Nasal cavity	score 0
	80	M	Testis	score 0
**Activated B-cell**	56	M	Inguinal lymph node	score 3
	42	F	Palatine tonsil	score 2
	66	M	Testis	score 2
	47	M	Palatine tonsil	score 0
	66	F	Axillary LN	score 2
	55	M	Palatine tonsil	score 2
	62	M	Testis	score 2
	54	M	Inguinal LN	score 2
	67	M	Palatine tonsil	score 4
	74	M	Heart	score 0
	67	M	Supraclavicular LN	score 0
	50	M	Palatine tonsil	score 1
	70	M	Nasal cavity	score 2
	54	F	Intestine, appendix, omentum, LN	score 0
	72	M	Stomach, large intestine, LN	score 2
	48	M	Cervical LN	score 0
	75	F	Small intestine	score 3
	70	M	Small intestine, appendix	score 4
	84	F	Nasal cavity	score 1
	70	M	Testis	score 2
**T-cell/histiocyte–Rich Large B-cell Lymphoma**	72	M	Neck, axillary LN	score 2
**Primary DLBCL of the CNS**	79	F	Brain	score 0
	78	F	Brain	score 0
	60	F	Brain	score 0
	78	M	Brain	score 0
**Primary Cutaneous DLBCL, Leg type**	77	F	Skin, upper arm and low leg	score 4
**EBV-Positive DLBCL, NOS**	62	M	Cervical LN	score 4
	86	F	Axillary LN	score 2

* LN: lymph node, DLBCL: diffuse large B-cell lymphoma, CNS: central nervous system, EBV: Epstein-Barr virus, NOS: not otherwise specified.

**Table 3 diagnostics-10-00275-t003:** The clinicopathologic characteristics of low IDO expression B-cell lymphomas.

Histologic Type	Age	Sex	Site of Involvement	IDO
Follicular Lymphoma				
Grade 1–2	44	M	Axillary LN	score 0
	79	M	Inguinal LN	score 0
	61	F	Cervical LN	score 0
	40	F	Inguinal LN	score 0
	51	F	Inguinal LN	score 0
	36	M	Abdominal LN	score 0
	62	F	Stomach	score 0
	50	M	Thoracic LN	score 0
	59	M	Cervical LN	score 0
	36	F	Cervical LN	score 0
	49	F	Buccal mucosa	score 0
Grade 3A	40	M	Cervical LN	score 0
	49	F	Epitrochlear LN	score 0
	77	M	Inguinal LN	score 0
Grade 3B	51	M	Submandibular LN	Diffuse area: score 3
**Pediatric Follicular Lymphoma**	32	M	Cervical LN	score 0
**MALT** **(mucosa-associated lymphoid tissue)-** **Lymphoma**	80	M	Eyelid	score 0
	56	F	Urinary bladder	score 0
	66	F	Stomach	score 0
	44	F	Stomach	score 0
	53	M	Eyelid	score 0
	65	M	Stomach	score 0
	71	M	Small intestine, omentum, LN	score 0
	57	F	Stomach	score 0
	81	M	Stomach	score 0
	63	M	Stomach	score 0
	42	F	Stomach	score 0
	66	M	Eyelid	score 0
	84	F	Stomach	score 0
	82	M	Stomach	score 0
	56	F	Stomach	score 0
	60	F	Rectum	score 0
**Nodal Marginal Zone Lymphoma**	66	F	Axillary LN	score 0
	66	F	Cervical LN	score 0
	67	F	Inguinal LN	score 0
	17	F	Palatine tonsil	score 0
**Mantle Cell Lymphoma**	71	M	Axillary LN	score 0
	53	M	Palatine tonsil	score 0
	66	F	Palatine tonsil	score 0
**Burkitt Lymphoma**	75	M	Stomach, pancreas	score 0
	47	M	Small intestine, appendix, omentum	score 0
**Chronic Lymphocytic Leukemia/Small Lymphocytic lymphoma**	62	M	Cervical LN	score 0
	71	M	Inguinal LN	score 0
**B-Lymphoblastic Lymphoma/Leukemia, NOS**	45	M	Soft tissue, paravertebral area	score 0

**Table 4 diagnostics-10-00275-t004:** The clinicopathologic characteristics and IDO expression in mature T- and NK-cell neoplasms.

Histologic Type	Age	Sex	Site of Involvement	IDO
**Extranodal NK-/T-cell Lymphoma**	65	F	Nasal cavity	score 4
	44	M	Nasal cavity	score 4
	26	F	Nasal cavity	score 0
	19	M	Scrotum	score 0
	83	M	Pharynx	score 4
	63	F	Nasal cavity	score 4
	64	M	Soft tissue, hip	score 4
	56	F	Nasal cavity	score 4
	58	M	Nasal cavity	score 4
	76	F	Nasal cavity	score 4
	43	M	Nasal cavity	score 4
	53	M	Nasal cavity	score 4
**Primary Cutaneous CD30 positive T-cell Proliferative Disorder**				
**Lymphomatoid Papulosis**	42	M	Skin, forearm and thigh	score 4
	57	M	Skin, occipital, and abdomen	score 0
	62	M	Eyelid	score 4
	40	M	Skin, shoulder, and flank	score 1
	62	M	Skin, thigh	score 1
	64	M	Skin, chest	score 1
**Primary Cutaneous Anaplastic Large Cell Lymphoma**	63	M	Skin, shoulder	score 3
	53	F	Skin, lower leg	score 0
**Peripheral T-cell Lymphoma, NOS**	78	F	Skin, thigh, and inguinal LN	score 3
	72	M	Axillary LN	score 3
	61	M	Skin, nose	score 1
	74	M	Axillary LN	score 3
	70	F	Cervical LN	score 3
	58	M	Cervical LN	score 3
**Anaplastic Large Cell Lymphoma, ALK-positive**	14	M	Skin and soft tissue, face	score 3
	21	F	Inguinal LN	score 4
	36	M	Soft tissue and bone, sternum	score 2
**Anaplastic Large Cell Lymphoma, ALK-negative**	71	M	Sphenoid sinus	score 0
	86	M	Vertebra, lumbar	score 0
**Primary Cutaneous CD8-positive Aggressive**				
**Epidermotrophic Cytotoxic T-cell Lymphoma**	82	M	Skin, abdomen	score 4
**Enteropathy-associated T-cell Lymphoma, Type II**	81	F	Small intestine	score 0
**Angioimmunoblastic T-cell Lymphoma**	55	M	Cervical LN	score 1
**Subcutaneous Panniculitis-like T-cell Lymphoma**	21	M	Skin, flank	score 4

* ALK: anaplastic lymphoma kinase.

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
