# Peer review of "Immunohistochemical Features of Indoleamine 2,3-Dioxygenase (IDO) in Various Types of Lymphoma: A Single Center Experience"

_diagnostics, 2020, doi:10.3390/diagnostics10050275_

Round 1

Reviewer 1 Report

This interesting study aims to investigate the rate and pattern of IDO expression in various types of lymphomas, scoring the immunohistochemical reactivity of neoplastic cells. The article is well-organized with well developed sections and easy to understand.

The introduction is properly concise. 

The second section (materials and methods) includes two subsections, both of them entitled "2.1. Study Population" and it seems to be a misprint.

About the population enrolled in the study it could be useful to make clear if the tissues were collected at the time of initial diagnosis or if there are samples obtained from patients previously treated with chemotherapeutic drugs.

Consider the possibility to add some clinical data including Ann Arbor stages, HIV status, EBV status.

With regard to the IDO expression via IHC it could be helpful to point out the staining pattern observed (i.e. cytoplasmic and/or nuclear staining) and highlight potential difficulties in assessing the percentage of tumor cell staining with particular reference to those lymphomas with abundance of reactive background cells (not only Hodgkin Lymphoma). Is it your opinion that a double IHC could help to identify the lineage of IDO producing cells?

The article includes fine pictures. Consider the possibility to attach photos with a higher magnification for lymphomas with low density of neoplastic cells.

Reviewer 2 Report

I appreciate your work, it's important for development of knowledge e for impact on therapeutical consequences.

I have some suggestions:

  1. it is necessary to change the number and title of second paragraph of "Materials and methods";
  2. it would be helpful if you could describe in "methods" protocol for in tumor cells and vessels detection of IDO (see Table 1, Hodgkin lymphoma);
  3. I believe you should comment  the meaning of your findings in different hystopathological entities in "discussion" section

Author Response

Response to Reviewer 2 Comments

Thank you for your conducive and kind advices.

Point 1: it is necessary to change the number and title of second paragraph of "Materials and methods";

Response 1: Sorry to bother you.  I fixed what you pointed out. (See below please)

2.1. Study Population 2.1. Study Population

-> 2.1. Study Population 2.2. Immunohistochemistry for Indoleamine 2, 3-dioxygenase

Point 2: it would be helpful if you could describe in "methods" protocol for in tumor cells and vessels detection of IDO (see Table 1, Hodgkin lymphoma);

Response 2: I added what you asked for. (See below please)

2.2. Immunohistochemistry for Indoleamine 2, 3-dioxygenase

…………..All lymphomas were scored based on percentage of tumor cell staining: 0 =<5%, 1 =5%–25%, 2 = 26%–50%, 3 =51%–75%, and 4 = >75 %. The moderate to strong cytoplasmic staining were regards as positive. In Hodgkin lymphomas, due to the scant cellularity of tumor cells, the expressions were evaluated to be positive or negative. The positivity for IDO in the vessels was judged in comparison with H&E stain. The vessels were lined by endothelium and covered by pericytes in H&E stain.

Point 3: I believe you should comment the meaning of your findings in different histopathological entities in "discussion" section

Response 3: I added what you asked for. (See below please) I'm sorry if there weren't enough answers. There have been few recent immunohistochemical assays of IDO in lymphomas. I hope that you would understand me.

  1. Discussion

All lymphomas were scored based on percentage of tumor cell staining: score 0 =<5%, 1 =5%–25%, 2 = 26%–50%, 3 =51%–75%, and 4 = >75%. It was difficult or impossible to distinguish reactive immune cells from tumor cells. To clarify the distinction between the tumor cells and reactive cells, multiple immunofluorescence stainings against IDO and other antibodies can be helpful as in the previous study [18]. The staining distribution of IDO was scored 0–4. A total of 80% (28/35) of mature T- and NK-cell neoplasms showed positivity for IDO protein (score 1: 5, score 2: 1, score 3: 7, score 4: 15). There was no different staining pattern or positive rate between the histopathological types of mature T- and NK-cell neoplasms. A total of 78.6% (22/28) of positive cases were scored 3 or 4; 83.3% (10/12) of extranodal NK/T cell lymphomas showed diffuse positivity for IDO protein. The remaining two negative cases were of relatively younger age (19 and 26 years) than the positive extranodal NK/T cell lymphomas (aged 43–83). In anaplastic large cell lymphomas, all three ALK-positive cases were positive (scoring 2, 3, or 4) for IDO and both ALK-negative cases were negative for IDO. A total of six peripheral T-cell lymphomas, NOS showed positivity for IDO (five scoring 3 and one scoring 1). In contrast to mature T- and NK-cell neoplasms, 29.9% (23/77) of mature B-cell lymphomas showed positivity for IDO protein (score 1: 3, score 2: 12, score 3: 4, score 4: 4). In mature B-cell lymphomas, 95.7% (22/23) of IDO positive cases were DLBCL, NOS or DLBCL, subtypes. There was no different staining pattern or positive rate between DLBCL, NOS and DLBCL, subtypes. All small B-cell neoplasms except one were negative for IDO protein. The remaining positive case was follicular lymphoma, grade 3B. The positive follicular lymphoma scored 3 at centroblasts in follicle and diffuse areas. In contrast to other subtypes of DLBCL, all four primary DLBCLs of the CNS were negative for IDO protein. In DLBCL, NOS

IDO proteins have been overexpressed in various cancers [22-25] and several IDO inhibitors have been assessed in multiple clinical trials. If an IDO inhibitor is to be commercialized, IDO immunohistochemistry will be important method. However, there are insufficient studies of immunohistochemistry for IDO in lymphomas. The previous studies had demonstrated only about Hodgkin lymphoma and diffuse large B cell lymphoma [15-17]. There was no immunohistochemical study of IDO protein for mature T- and NK-cell neoplasms. Our study included various types of mature T- and NK-cell lymphoma that had previously been unreported and showed various patterns of IDO staining according to the type. Our study used scant sample size to determine the IDO pattern of various lymphomas and the relationship between IDO expression and prognosis. When the results are accumulated, IDO immunohistochemistry will be a useful tool to diagnose lymphomas and predict their prognosis.